# Criterion Validity of Linear Accelerations Measured with Low-Sampling-Frequency Accelerometers during Overground Walking in Elderly Patients with Knee Osteoarthritis

**DOI:** 10.3390/s22145289

**Published:** 2022-07-15

**Authors:** Arash Ghaffari, Ole Rahbek, Rikke Emilie Kildahl Lauritsen, Andreas Kappel, Søren Kold, John Rasmussen

**Affiliations:** 1Interdisciplinary Orthopaedics, Aalborg University Hospital, 9000 Aalborg, Denmark; o.rahbek@rn.dk (O.R.); r.lauritsen@rn.dk (R.E.K.L.); andreas.kappel@rn.dk (A.K.); sovk@rn.dk (S.K.); 2Department of Materials and Production, Aalborg University, 9220 Aalborg East, Denmark; jr@mp.aau.dk

**Keywords:** inertial measurement units, wearable motion-tracking sensors, low-sampling-frequency accelerometers, knee osteoarthritis, SENS sensors, remote monitoring of patients, gait accelerations, time-domain comparison, frequency-domain comparison, test–retest reliability

## Abstract

Sensors with a higher sampling rate produce higher-quality data. However, for more extended periods of data acquisition, as in the continuous monitoring of patients, the handling of the generated big data becomes increasingly complicated. This study aimed to determine the validity and reliability of low-sampling-frequency accelerometer (SENS) measurements in patients with knee osteoarthritis. Data were collected simultaneously using SENS and a previously validated sensor (Xsens) during two repetitions of overground walking. The processed acceleration signals were compared with respect to different coordinate axes to determine the test–retest reliability and the agreement between the two systems in the time and frequency domains. In total, 44 participants were included. With respect to different axes, the interclass correlation coefficient for the repeatability of SENS measurements was [0.93–0.96]. The concordance correlation coefficients for the two systems’ agreement were [0.81–0.91] in the time domain and [0.43–0.99] in the frequency domain. The absolute biases estimated by the Bland–Altman method were [0.0005–0.008] in the time domain and [0–0.008] in the frequency domain. Low-sampling-frequency accelerometers can provide relatively valid data for measuring the gait accelerations in patients with knee osteoarthritis and can be used in the future for remote patient monitoring.

## 1. Introduction

Knee osteoarthritis (OA) is a prevalent musculoskeletal condition affecting the older population, characterized by articular-cartilage degeneration and joint-space narrowing. Pain and walking problems associated with knee OA affect patients’ quality of life and impose a considerable economic burden on the health care system [1]. A timely follow-up of patients with knee OA is required to monitor the outcome of the applied treatment and provide patients with appropriate rehabilitation programs, exercises, and education [2,3]. However, frequent outpatient visits can be associated with lower patient satisfaction [4,5]. Furthermore, considerable costs and inherent difficulties with transport and situations such as the COVID-19 pandemic impel follow-up visits to shift from routine clinical visits to the remote home monitoring of patients.

In addition to questionnaires and patient-reported outcome measures (PROMs), wearable motion-tracking sensors, such as Inertial Measurement Units (IMUs), have recently been introduced in telemedicine programs. These wearable sensors provide a practical and cost-effective solution for obtaining objective functional measures regarding the characteristics of movements and physical activities [6,7,8]. Commercially available IMUs have a wide range of technical specifications based on their usage, such as the range of sensitivity, the number of axes, electrical characteristics, the interface, and the included sensor types. One of the most critical features of wearable sensors for patient monitoring is battery lifetime. Long battery life without recharging in a small sensor would usually require the IMU to comprise only passive accelerometers (as opposed to active gyroscopes) and to operate at a low sampling frequency. Most modern IMUs in human-motion-capture technology sample at frequencies between 10 and 200 Hz [9]. Different sampling rates have been recommended for capturing daily life activities depending on the type of activities and the type of analysis required [10]. Previous investigations of the SENS sensor [11,12] have focused on activity classification. Still, we could not find a study evaluating the accuracy and precision of the measurements and how a low sampling frequency might compromise them. The ultimate goal of studies of IMUs in gait analysis is to examine their clinical applicability; however, the usefulness of a measure in a clinical setting depends on the extent to which clinicians can rely on the data being accurate and precise [13].

The main aim of our study was to validate the measurements performed by a commercially available sensor (SENS Motion^®^; Copenhagen, Denmark) designed for measuring physical activities in the healthcare sector and for research projects. We investigated whether the accelerations measured with small low-frequency accelerometers in patients with knee OA were valid representations of the true accelerations for kinematic gait assessment. We compared the accelerations measured with SENS sensors at ~12.5 Hz with a standard inertial-sensor-based motion-capture system as the gold standard to determine the criterion validity of accelerometers with low sampling frequencies. We also evaluated the test–retest reliability of the measurements obtained with low-sampling-frequency sensors.

The key contributions of this study can be summarized as follows:The accuracy and precision of the measurements of low-sampling-frequency accelerometers were determined to evaluate the performance of these sensors in the time and frequency domains and with respect to different coordinate axes;By employing musculoskeletal modeling, we could compare the accelerations of two systems at an exact location. We could compare the accelerations in close-to-real-life situations and without needing sophisticated motion and gait laboratories;In addition, we evaluated the test–retest repeatability of the accelerations measured with these sensors and compared them with the criterion system.

## 2. Materials and Methods

### 2.1. Participants

The study participants consisted of patients at different stages of unilateral knee OA referred to Aalborg University Hospital, Denmark, between December 2021 and March 2022 and diagnosed by specialists in orthopedic surgery. We also included asymptomatic individuals of over 50 years of age at stage 0 of knee OA. Participants complaining of pain and discomfort in the spine and other lower-limb joints, and patients with high BMIs (>35 kg/m^2^), a recent history of operations in the lower limbs, neurological movement disorders, and inflammatory arthritis were excluded from the study. The study was approved by Regional Committee on Health Research Ethics (journal number 2021-000438).

### 2.2. Equipment

Two sensors were used in this study (Table 1). Figure 1A simultaneously demonstrates the application of both sensors.

#### 2.2.1. Xsens

The Xsens sensors (Xsens, Enschede, The Netherlands) formed the MVN Awinda wireless full-body MoCap system. They consisted of 17 IMUs, which comprised a 3D gyroscope, a 3D accelerometer, and a 3D magnetometer combined with sensor fusion algorithms. The sensors were placed on the head, shoulders, upper arms, forearms, sternum, pelvis, upper and lower legs, and feet (Figure 1A) for full-body motion capture. This system has been repeatedly studied against optical motion-capture systems for gait analysis. Fair–excellent inter-rate and intra-rate reliability and excellent system validity have been demonstrated [14,15,16,17,18,19]. The Xsens system’s bias rate is comparable to the gold-standard camera-based motion-capture gait analysis [19]. The high inter-rater reliability of the Xsens system, supporting its stability and independence of the rater, and its high validity against gold-standard gait-analysis methods made it a good choice for the criterion method in our study.

#### 2.2.2. SENS

SENS sensors (SENS Motion^®^, Copenhagen, Denmark) only contain a 3D accelerometer. These sensors are medically approved devices designed for the long-term monitoring of patients’ physical activities. In addition, they have an internal memory for storing up to 14 days of data and cloud connectivity via a mobile telephone. Following the manufacturer’s instructions, we attached two SENS sensors on the lateral distal side of each thigh about 10 cm above the lateral femoral epicondyle (Figure 1B).

### 2.3. Data Collection

The experiment process was explained to the eligible participants, and informed consent was obtained before inclusion. Afterward, the subjects’ basic information was registered, including their age, sex, weight, height, and the severity of knee OA according to the Kellgren–Lawrence classification [20]. The Knee injury and Osteoarthritis Outcome Score (KOOS) [21] was also obtained as a subjective score of the patient’s problems regarding knee OA.

Gait data from SENS and Xsens sensors were simultaneously recorded by having patients perform two overground walking trials at a self-selected speed. Since patients with knee OA might demonstrate asymmetries in knee biomechanics during walking [22], we used two SENS sensors for each participant (one on each thigh) to separately compare each side. The time interval between the two trials was 2–5 min long. Each walking trial consisted of a walk in a straight line across the outpatient clinic (approximately 15 m). Data from Xsens sensors were recorded on a local computer with MVN Analyze Software. Data from the SENS sensors were saved in the onboard memories of the sensors and transferred to a cloud storage system, from which we downloaded and processed the data.

### 2.4. Data Processing

The gait data from the Xsens sensors were processed with MVN Analyze software (version 2021.2; Xsens, Enschede, The Netherlands). Afterward, the processed data were used as inputs for AnyBody Modeling System (version 7.3.4; AnyBody Technology, Aalborg, Denmark). AnyBody is musculoskeletal modeling software that enables the simulation and analysis of the kinematics and kinetics of human movements to be performed. Xsens relies on the sensor fusion of a multitude of channels to recover positional information, and the AnyBody Modeling System further processes the signals by taking the kinematic constraints into account before returning the local accelerations. This pipeline, to some extent, compensates for the influence of soft-tissue artifacts on the local accelerations.

A musculoskeletal model for each participant was created, and two virtual representations of the accelerometers (VirtualSENS) were defined in each model on the right and left thighs at the locations of the two SENS sensors. The model accepted the processed data from MVN Analyze as inputs. The outputs consisted of the 3D linear accelerations offset by gravity in the local coordinate systems of the VirtualSENS sensors, which we employed as the criterion to validate the SENS signals. For convenience, we refer to the signals obtained with VirtualSENS as Criterion signals in this article. 

The gait data obtained from the SENS and VirtualSENS sensors consisted of comparable linear accelerations acc_x_, acc_y_, and acc_z_ along the three perpendicular local axes, x, y, and z, respectively (Figure 1C). In addition, the length of the acceleration vector, n, was also calculated.

Each patient generated 16 Criterion and 16 SENS signals during the two trials, considering two sides (left and right sensors) and four axes (x, y, z, and n).

We inspected the frequency contents of both signals to determine a cutoff frequency and performed a residual analysis of the difference between filtered and unfiltered signals over a wide range of cutoff frequencies [23]. A fourth-order zero-lag low-pass Butterworth filter with a cutoff frequency of 4 Hz was used. 

AnyBody interpolated the high-frequency data from Xsens, resulting in a continuous Criterion signal that can be resampled at any frequency. The Fourier method was used to reconstruct a continuous interpolation of the SENS signals drawing upon their periodic nature to improve the bridging between samples (Figure 2).

With the SENS system measuring continuously, there was no simple way to establish a mutual time stamp for the SENS and Xsens signals. Given that the participants were elderly and suffering from knee pain, they could not be required to perform a fast move or a jump to synchronize the two systems. Cross-correlation was, therefore, used to synchronize a sequence of five gait cycles within the two signals in the middle of the trial, and this appeared to be reliable, given the inevitable deviations between the steps in a sequence. The acceleration signals in the vertical direction (*x*-axis) were used as the reference axis for synchronization. The resulting time stamps were employed for the signals along the other axes. The five synchronized cycles were subsequently segmented into five individual cycles using autocorrelation as described by Yang et al. [24]. An average cycle was computed and linearly normalized to the percent gait cycle. Signal processing was conducted with Python (Python Software Foundation; Python Language Reference; version 3.10; available at http://www.python.org (accessed on 15 January 2022)).

### 2.5. Statistical Analysis

Descriptive statistics were used to quantify the characteristics of the participants. Anthropometric data and gait cadence were presented as means and standard deviations, age was described with median and range, and knee OA scores were reported as median and interquartile range. To compare the continuous Criterion and SENS signals, we discretized the signals of the average gait cycles by resampling them at the rate of 60 Hz. Kolmogorov–Smirnov tests were applied to check for the assumption of normal distribution (*p* < 0.05). For the statistical analyses, we used the R statistical package, version 4.1.0 (RStudio Team (2021); RStudio: Integrated Development for R. RStudio, PBC, Boston, MA, USA; URL http://www.rstudio.com/ (accessed on 8 May 2022)).

#### 2.5.1. Test–Retest Reliability

Interclass correlation coefficient (ICC) estimates and 95% confidence intervals were calculated for the measured accelerations in the averaged gait cycles between the first and second trials with respect to different coordinate axes and the vector magnitude. The ICCs for both SENS and Criterion signals were calculated by employing two-way mixed effects, absolute agreement, and the single-rater model.

#### 2.5.2. Time-Domain Comparison

The synchronized SENS and Criterion time series were visually assessed to determine the agreement between the signals in the time domain. The scatterplots for the correlations between the SENS and Criterion measurements were generated. The concordance correlation coefficient (CCC) introduced by Lin [25] was employed to evaluate the agreement between the SENS and Criterion accelerations measured at each time point of the averaged gait cycle with respect to different coordinate axes. Since Kolmogorov–Smirnov tests showed skewness and long tails in the distribution of the measurements, the non-parametric estimation of the CCC was applied [26]. 

Regarding the absolute agreement, the method described by Bland and Altman [27] was employed to calculate the limits of agreement (LoAs) between the SENS and Criterion measurements. Due to the non-normal distribution of the acceleration differences, the LoA was defined as the interval containing 95% of the differences; the lower and upper LoAs corresponded to the 2.5th and 97.5th percentiles, respectively [28]. The deviation of the median of the differences (50th percentiles) from zero was considered the bias in the measurement. The 95% confidence intervals for the upper and lower LoAs and the bias were estimated with bootstrapping. Spearman’s rho (r_s_) was calculated for the correlations between the differences and Criterion to demonstrate the stability of the bias across the range of Criterion values.

The dynamic time warping (DTW) algorithm [29] was used to find the alignment between two signals (SENS and Criterion averaged gait cycles). A normalized similarity index (*NSI*) [30] was then calculated using the following equation:$$NSI=\frac{M\left(\mathrm{Criterion}\right)-d\left(\mathrm{Criterion},\;\mathrm{SENS}\right)}{M\left(\mathrm{Criterion}\right)}$$
where *d*(Criterion, SENS) is the distance between the SENS and Criterion signals, measured with the DTW method, and *M* (Criterion) is calculated by multiplying the number of the samples in the Criterion signals by the range of the acceleration values of Criterion (*Max_acc_* – *min_acc_*). 

#### 2.5.3. Frequency-Domain Comparison

To compare the signals in the frequency domain, we performed the Fast Fourier Transform and evaluated the frequency contents of the signals. After generating the signals’ power spectral densities (PSDs), the frequencies of the peaks and the powers corresponding to those frequencies were compared between the SENS and Criterion signals. Scatterplots and Bland–Altman plots were sketched, and non-parametric LoAs and CCCs were calculated to evaluate the agreement between the SENS and Criterion signals regarding the frequencies and the powers of the PSD peaks.

Subsequently, the Fourier series representation of the signals was generated using the following equations as described by Skejø et al. [31]:$$y\left(t\right)=a_{0}+\sum_{i=1}^{n}a_{i}\cos\left(i\omega t\right)+\sum_{i=1}^{n}b_{i}\sin\left(i\omega t\right)$$
$$a_{0}=\frac{1}{T}\;\int^{\;}f\left(t\right)dt$$
$$a_{i}=\frac{2}{T}\int_{0}^{T}f\left(t\right)\cos\left(\omega it\right)dt$$
$$b_{i}=\frac{2}{T}\int_{0}^{T}f\left(t\right)\sin\left(\omega it\right)dt$$
where $a_{0}$, $a_{i}$, and $b_{i}$ are Fourier coefficients, n is the number of Fourier-coefficient pairs, ω is the angular stride frequency, and *T* is the cycle time computed as $T=\frac{2\pi }{\omega }$. The first ten pairs of Fourier coefficients ($a_{0},\;a_{1},b_{1},\ldots ,\;a_{10},\;b_{10}$) of the SENS and Criterion signals were compared, and non-parametric Bland–Altman LoAs and CCCs were calculated.

An overview of the experimental protocol used in this study is depicted in Figure 3.

## 3. Results

### 3.1. Participants

We included 44 participants (25 females) at different stages of unilateral knee OA (stages 0–IV). The characteristics of the subjects are demonstrated in Table 2. Overall, we evaluated 880 gait cycles of right and left legs.

### 3.2. Test–Retest Reliability

The result of the test–retest reliability evaluation showed an excellent repeatability of the acceleration measurements obtained with SENS between the first and the second trials. The calculated ICCs for SENS were comparable with those for Criterion (Table 3). 

### 3.3. Time-Domain Comparison

The scatterplots of the accelerations measured with SENS and Criterion demonstrated a high correlation between the measurements. However, a slight gradual drift from the line of equality was observed as the accelerations increased or decreased from −1 on the *x*-axis and 0 on the *y*- and *z*-axes (Figure 4). Furthermore, with respect to the magnitude of the measured accelerations (*n*), a relative widening of the SENS accelerations was observed for higher Criterion accelerations. 

The corresponding Bland–Altman plots for the agreement between the measured accelerations obtained with SENS and Criterion demonstrated that the differences between the measurements altered with the changes in the Criterion accelerations and confirmed the above-mentioned findings. The correlations between the Criterion–SENS acceleration differences and Criterion, as estimated by Spearman’s rank correlation, were 0.13, 0.34, −0.14, and 0.07 with respect to the *x-, y-,* and *z*-axes, and the vector magnitude (*n*), respectively (Figure 4).

The calculation of the CCC demonstrated a higher agreement between SENS and Criterion with respect to the *x*- and *y*-axes and the magnitude of the accelerations (*n*) than with respect to the *z*-axis (Table 4). The result of the NSI showed a slightly higher similarity with respect to the *y*-axis. The results of the right and left sides were not significantly different.

The absolute differences in the accelerations obtained with SENS and Criterion were more evident with respect to the *y*-axis, as demonstrated by the range of the LoAs being broader than that with respect to the other axes (Table 5). However, the calculated bias with respect to all axes was trivial. Similar results for the right and left sides were observed.

### 3.4. Frequency-Domain Comparison

A high correlation was found in the scatterplots between the frequencies of the PSD peaks of the SENS and Criterion signals (Figure 5). The Bland–Altman plots also demonstrated good agreement, especially at lower frequencies, as the differences increased at higher frequencies. Spearman’s correlations between Criterion–SENS frequency differences and the Criterion frequency for both sides were 0.16, 0.14, 0.09, and 0.25 with respect to the *x*-, *y*-, and *z*-axes, and the vector magnitude (*n*), respectively (Figure 5).

The correlation and the agreement observed in the scatterplots between the powers of the PSD peaks of the SENS and Criterion signals were inferior compared with the peak frequencies, especially with respect to the *z*-axis (Figure 6).

The scatterplots and Bland–Altman plots in Figure 7 indicated high correlation and agreement between the Fourier coefficients of SENS and Criterion with respect to the three coordinate axes and the magnitude of the accelerations (*n*). Both sides’ r_s_ were −0.07, 0.15, 0.23, and 0.06 with respect to the *x*-, *y*-, and *z*-axes, and the accelerations’ magnitude (*n*), respectively.

The CCC of the agreement between SENS and Criterion was excellent regarding the value of the frequencies of the PSDs and the Fourier coefficients; however, it was more variable among the different axes with respect to the powers of peaks of the PSDs (Table 6).

The absolute agreement of the frequencies and powers of the PSD peaks, and the Fourier coefficients of SENS and Criterion are provided in Table 7 by with the upper and lower LoAs and the bias estimated with the Bland–Altman method. 

## 4. Discussion

This study evaluated the validity and reliability of the linear accelerations measured with low-sampling-frequency (~12.5 Hz) accelerometers in elderly individuals with and without knee OA during simple overground walking. The results demonstrate that in this group of patients, after the same signal processing, the measurements obtained with these accelerometers were highly correlated with previously validated inertial-based motion-capture gait analysis in both the time and frequency domains. In addition, the high repeatability of the measurements of these accelerometers was demonstrated.

IMUs with a wide range of characteristics are used for human movement analysis, and one of the most determining characteristics of an IMU is its sampling rate. A higher sampling frequency creates higher-quality data but comes at a price. As the rate of sampling increases, more data must be handled, stored, and transferred, which would challenge the continuous monitoring of patients for more extended periods. Moreover, a higher sampling frequency results in higher energy consumption and shorter battery lifetime and requires more expensive, complex, and bulky sensors. Accordingly, several AI algorithms have recently been proposed to increase the accuracy of the low sampling frequency of IMUs [32,33]. It has been demonstrated by previous studies that 98% of the walking signal’s power is below 10 Hz [34] and significantly below 5 Hz [35]. In contrast with physical activities such as sports and dancing, which might require higher sampling frequencies, simple gait in the elderly is relatively slow and contains lower frequencies that theoretically enable the capturing of data to be conducted with more simplistic sensors. However, there is a lack of evidence on whether IMUs with low sampling rates can provide accurate and precise data for evaluating human movements, especially activities containing lower frequencies.

Measurements are never entirely free from errors. Random errors or noise can affect the precision of the measurements or how reproducible the exact measurement is under similar circumstances, and systematic errors or biases affect the accuracy of measurements or how close the observed value is to the actual value. In this study, we evaluated the accuracy and precision of the measurements performed by a low-sampling-frequency accelerometer (SENS) by comparing the accelerations in the time and frequency domains against an IMU system that has been proved to have high reliability and validity in motion analysis. Our findings show that the accelerometer with a sampling frequency of ~12.5 Hz could adequately capture the gait acceleration signals, even though the cadence of the gait in our study participants was not low (110 ± 11 steps per minute). 

In this study, we determined the test–retest reliability to evaluate the consistency of the measurements across time and based on the guidelines for reporting ICCs in reliability research [36]; the test–retest results indicated excellent reliability between the first and second trials of the acceleration measurements obtained by the SENS sensors. The results were comparable with Criterion and also with previous studies [37,38]. We examined the test–retest reliability of the SENS sensor in the same session with a few-minute interval to avoid the influence of pain and fatigue on the stability of the gait pattern. In addition, given the high between-days variability in gait pattern and speed, especially in subjects with knee OA, we decided to capture the data on the same day [39].

In the time-domain comparison, the highest correlations were found with respect to the *x*- and *y*-axes (vertical and anteroposterior axes). The correlation with respect to the *z*-axis (mediolateral axis) was relatively lower. The magnitudes of the SENS and Criterion signals’ vectors demonstrated a relatively high correlation. DTW and correlation capture different aspects of similarity between two time series; nonetheless, the DTW method to calculate the distance between the compared signals was also in agreement with the correlation analysis. The lower correlation with respect to the mediolateral axis might be due to the higher frequency and lower amplitude of the movements on this axis, which might lead to an inferior performance of the sensors. Although the accelerations in the anteroposterior and vertical directions have higher magnitudes, there could also be essential data in the mediolateral direction, especially in pathologic gaits, and it should not be overlooked. 

The results of the absolute agreement between the measurements based on the Bland–Altman method demonstrated a slight bias in all directions. Several studies have provided information regarding the differences in the accelerations measured by inertial sensors attached to the thighs among patients with knee OA and asymptomatic control groups [40,41], and according to our analysis, these values with respect to different axes were within the range of 90% LoAs for SENS and Criterion. Other key findings of the analysis of the Bland–Altman plots of the acceleration measurements in the time domain were the narrower range of the LoAs and the lower variability of the differences in the magnitude of the SENS accelerations’ vector compared with the other three axes (*x*, *y*, and *z*), which implies that the magnitude of the accelerations might be more accurate and reliable than each of the individual perpendicular axes.

Comparing the frequencies of the signals also demonstrated near-perfect concordance between the frequencies of the peaks of the signals, which signifies that these sensors could reliably measure the periodicity and harmonics of the gait. However, as expected, due to the limits of SENS, the differences in the measured frequencies increased at higher frequencies. Regarding the powers of the peaks of the PSDs, the broader LoAs and the weak correlation coefficients with respect to the *z*-axis compared with the other axes suggest a higher noise content and an inferior capability of the sensors of capturing data entirely along the mediolateral axis. The comparison of the Fourier coefficients, however, demonstrated higher correlations, lower biases, smaller ranges of the LoAs, and more stable performances with respect to all coordinate axes, which suggests that it could be a more appropriate and reliable method for further signal analysis of data recorded with these low-sampling-frequency sensors.

Previous studies reporting the performance of low-sampling-frequency accelerometers have mainly focused on the sensors’ capability of performing physical-activity measurement and classification. To our knowledge, this is the first study evaluating the accuracy and precision of the acceleration measurements obtained with small low-sampling-frequency wearable sensors in the elderly population with and without knee OA. In addition, the assessment of the signals in both the time and frequency domains provides a basis for further studies of the capabilities of these sensors in kinematic gait analysis. The most prominent strength of our study was that by applying musculoskeletal modeling to process the Criterion signals, we could compare the linear accelerations at the exact location of the sensors, which allowed us to favorably compare the acceleration signals in an ordinary outpatient clinic environment, without using sophisticated gait laboratories. The most important limitation of our study was that we could not synchronize the signals during data recording. Since the participants were elderly and suffered from knee OA, we could not ask them to perform a fast move or a jump. Instead, we had to synchronize the signals during postprocessing using the cross-correlation method. 

The sensors’ sampling frequency of 12.5 Hz is insufficient for a comprehensive gait analysis but may suffice to detect abnormal gait and grade gait deviation. The essential prerequisite in this regard would be the capability of the sensors to measure the accelerations as accurately and reliably as allowed by the limitations of current technology, which was this study’s primary objective. The path is now open to exploring the capability of these sensors in remote gait-quality assessment in telemedicine. 

A detailed comparison of the signals in the time and frequency domains helps us to identify the most reliable and beneficial features of machine learning algorithms and interpret the data from wearable sensors in remote patient monitoring, similar to myoelectric biomarkers, which have been proposed in post-stroke gait [42]. This study demonstrated that, despite the low sampling frequency, SENS sensors could accurately measure lower-limb accelerations in elderly patients with and without knee OA, especially with respect to the anteroposterior and vertical axes. Furthermore, the measurements’ accuracy and precision were higher with respect to the Fourier coefficients of the signals. This finding indicates the usefulness of Fourier coefficients in the interpretation of data obtained with low-sampling-frequency sensors. Considering the underlying kinematic variations in knee OA patients [43,44,45], we could demonstrate that low-sampling-frequency accelerometers are capable of reliably measuring the accelerations in this population. Nevertheless, further studies are required to answer whether these sensors with low sampling frequencies can distinguish between normal and abnormal gaits due to knee OA and how accurate and trustworthy they can be.

## 5. Conclusions

In elderly patients with and without knee OA performing overground walking, low-sampling-frequency accelerometers can provide measurements with relatively high accuracy and precision, especially with respect to the anteroposterior and vertical axes. Furthermore, the validity and reliability of the measurements, especially with respect to the Fourier coefficients of the signals, indicate their applicability in telemedicine for the remote monitoring of patients. However, further studies are required to demonstrate their practicality for this purpose.

## Figures and Tables

**Figure 1 sensors-22-05289-f001:**
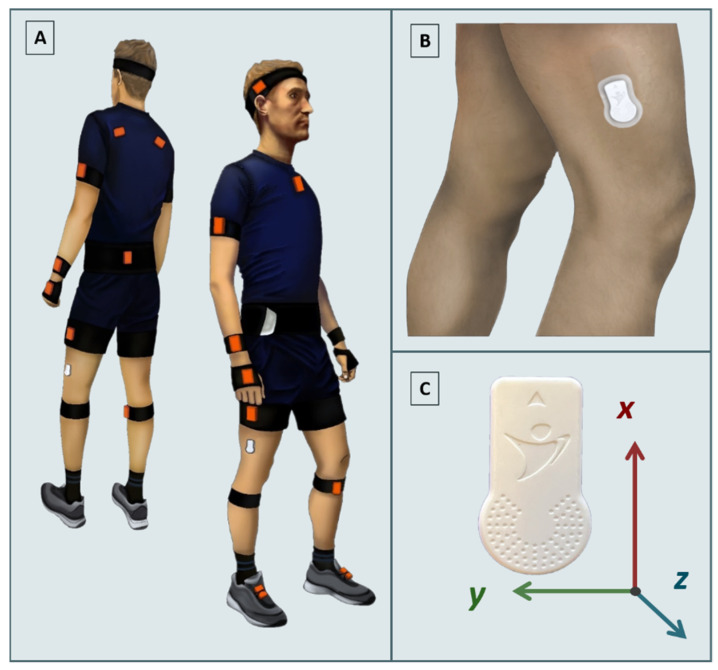
(**A**) Simultaneous application of SENS and Xsens sensors (Xsens sensors are shown on top of clothing for illustrative purposes). (**B**) Placement of a SENS sensor on the distal lateral side of the thigh. (**C**) Coordinate axes of the SENS sensor.

**Figure 2 sensors-22-05289-f002:**
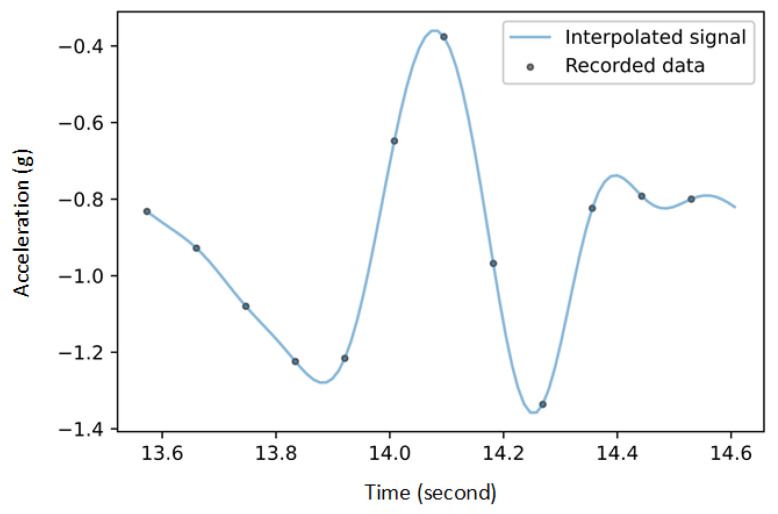
Reconstruction of continuous SENS signal using the Fourier method. The gray dots show the accelerations recorded by the SENS sensors along the x-axis as an example of one gait cycle. The blue line shows the continuous signal reconstructed using the Fourier method.

**Figure 3 sensors-22-05289-f003:**
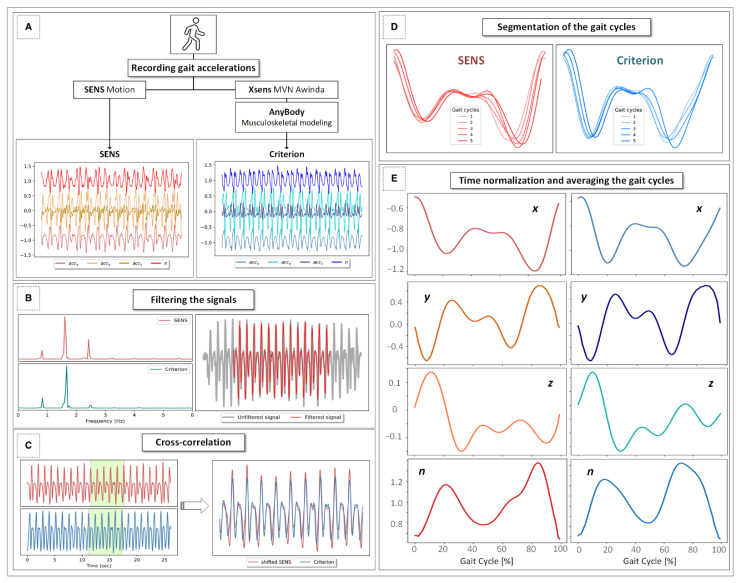
An overview of the signal processing protocol used in this study: (**A**) Simultaneous recording of the gait signals obtained with SENS and Xsens during two overground gait trials. (**B**) Inspection of the PSD (power spectral density) of the signals and filtering of the signals after determining a cutoff frequency of 4 Hz using a fourth-order zero-lag low-pass Butterworth filter. (**C**) Temporal matching of the signals using the cross-correlation method. (**D**) Segmentation of the gait into five individual gait cycles. (**E**) Averaging and normalization of the gait cycles into gait cycle percentages with respect to different coordinate axes and the magnitude vector.

**Figure 4 sensors-22-05289-f004:**
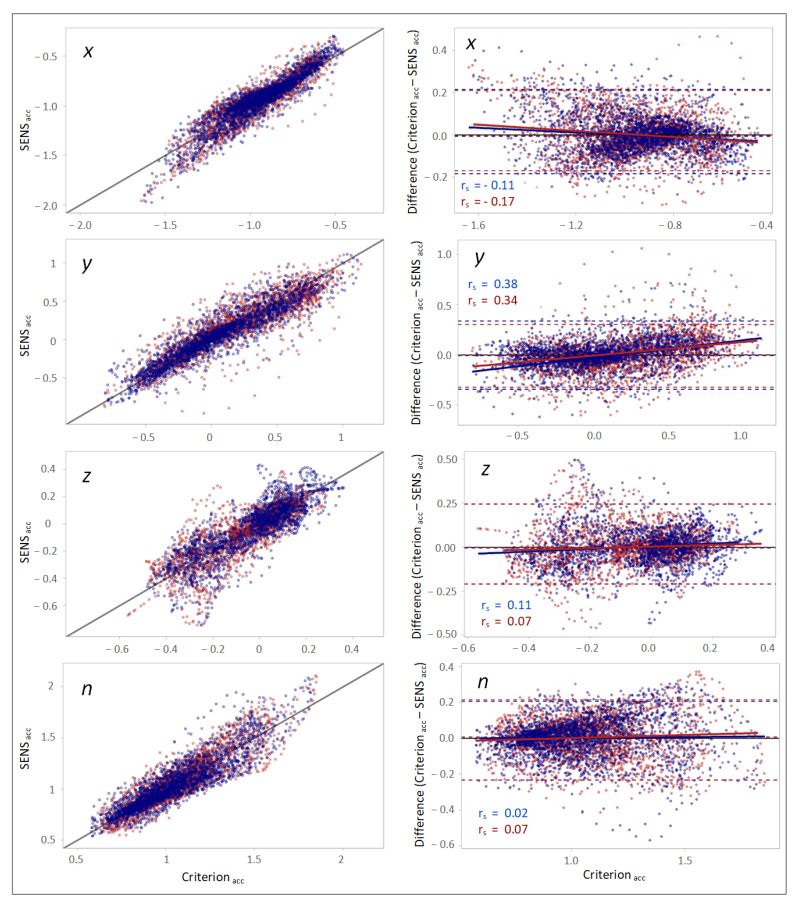
(**Left column**): scatterplots demonstrating the correlation between the SENS and Criterion accelerations with respect to different axes. The black line depicts the line of equality (SENS acc = Criterion acc). (**Right column**): Bland–Altman plots demonstrating the agreement between the SENS and Criterion accelerations. The upper and lower LoAs (limits of agreement) are shown as the upper and lower dashed lines corresponding to the 2.5th and 97.5th percentiles of the differences. (Data from the right-side sensor are marked in blue, and those from the left-side sensor is marked in red).

**Figure 5 sensors-22-05289-f005:**
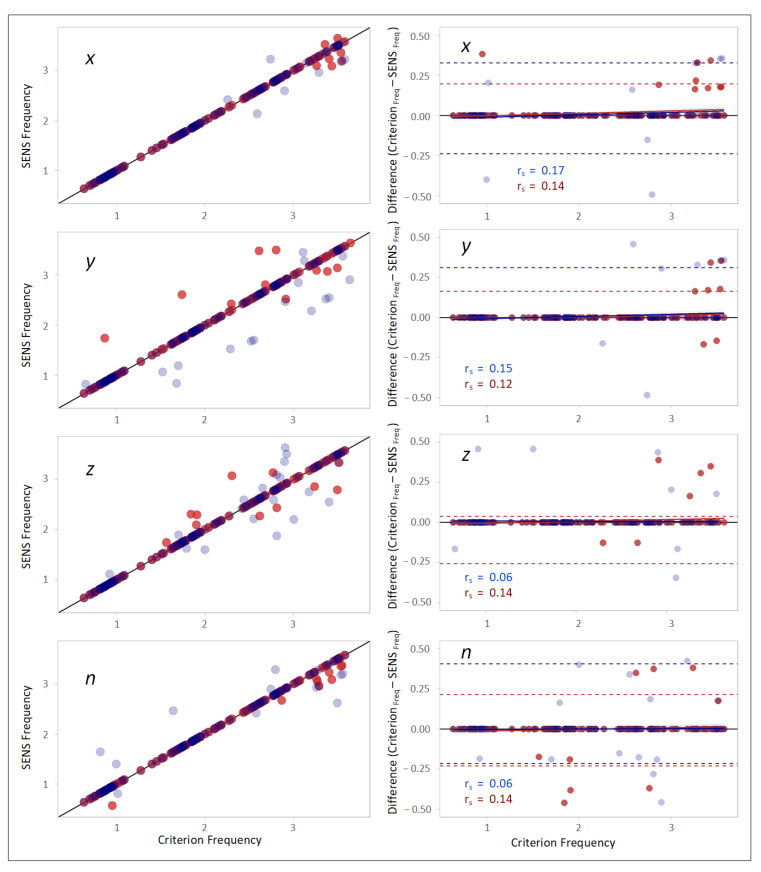
(**Left column**): scatterplots demonstrating the correlation between the frequencies of the peaks of the PSDs of SENS and Criterion with respect to different axes. The solid black line depicts the line of equality (SENS_Freq_ == Criterion_Freq_). (**Right column**): corresponding Bland–Altman plots demonstrating the agreement between the frequencies of the peaks of the PSDs of SENS and Criterion. The upper and lower LoAs are shown as the upper and lower dashed lines corresponding to the 2.5th and 97.5th percentiles of the differences. (Data from the right-side sensor are marked in blue, and those from the left-side sensor are marked in red).

**Figure 6 sensors-22-05289-f006:**
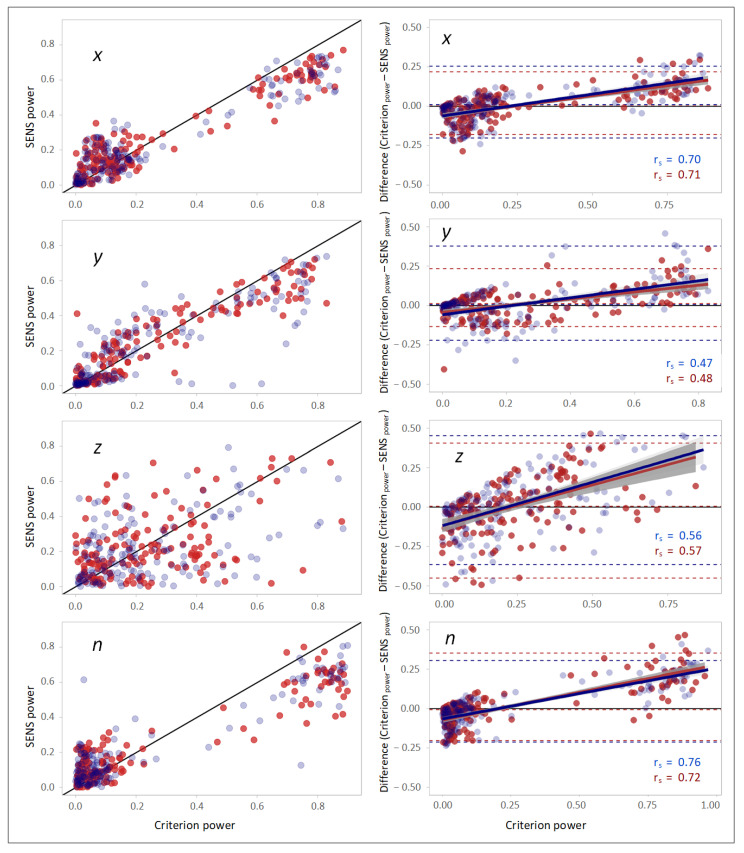
(**Left column**): scatterplots demonstrating the correlation between the powers of the peaks of the PSDs of SENS and Criterion with respect to different axes. The solid black line depicts the line of equality (SENS_power_ == Criterion_power_). (**Right column**): corresponding Bland–Altman plots demonstrating the agreement between the powers of the peaks of the PSDs of SENS and Criterion. The upper and lower LoAs are shown as the upper and lower dashed lines corresponding to the 2.5th and 97.5th percentiles of the differences. (Data from the right-side sensor are marked in blue, and those from the left-side sensor are marked in red).

**Figure 7 sensors-22-05289-f007:**
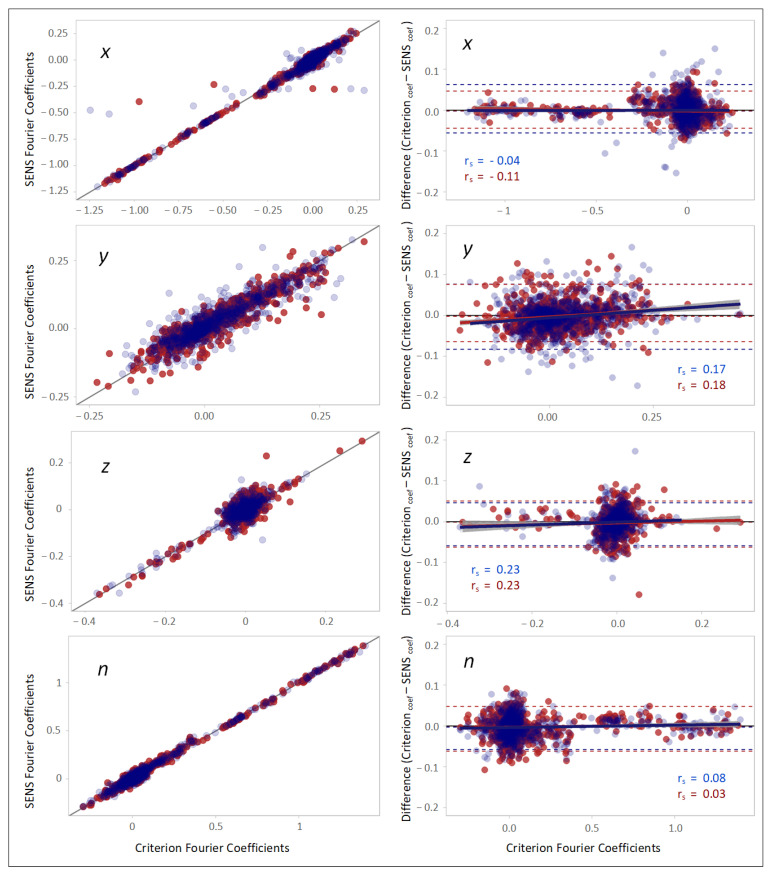
(**Left column**): scatterplots demonstrating the correlation between the Fourier coefficients of SENS and Criterion with respect to different axes. The solid black line depicts the line of equality (SENS coefficients == Criterion coefficients). (**Right column**): corresponding Bland–Altman plots demonstrating the agreement between the Fourier coefficients of the peaks of the PSDs of SENS and Criterion. The upper and lower LoAs are shown as the upper and lower dashed lines corresponding to the 2.5th and 97.5th percentiles of the differences. (Data from the right-side sensor are marked in blue, and those from the left-side sensor are marked in red).

**Table 1 sensors-22-05289-t001:** Technical specifications of the two types of sensors used in the study.

	SENS	Xsens
Dimension	47 × 22 × 4.5 mm	47 × 30 × 13 mm
Weight	7 gr	16 gr
Sampling frequency	12.5 Hz	60 Hz
3D accelerometer	±4 G	±16 G
Battery life	15 weeks	6 h
Attachment	3 M patches	Velcro straps

**Table 2 sensors-22-05289-t002:** Characteristics of the study participants.

Variable	Value
Sex (n (%))	
Female	25 (57)
Male	19 (43)
Median age (years) [range]	65.6 [48.1–85.4]
Height (cm)	172.8 ± 8.7
Weight (kg)	80.9 ± 14.3
BMI (kg/m^2^)	27.0 ± 3.7
Gait cadence (steps/min)	110 ± 11
Painful knee (n (%))	
Left	15 (34)
Right	17 (39)
No	12 (27)
Severity of knee OA ^1^ (n (%))	
0	12 (27.3)
I	4 (9.1)
II	9 (20.5)
III	13 (29.5)
IV	6 (13.6)
KOOS ^2^ Score	
Pain	62.5 [47.2–97.9]
Symptoms	67.9 [52.7–91.1]
ADL ^3^	69.1 [55.5–97.4]
Sport/Rec ^4^	32.5 [13.8–86.3]
QOL ^5^	40.6 [31.3–89.1]

^1^ Based on Kellgren–Lawrence classification. ^2^ Knee injury and Osteoarthritis Outcome Score. ^3^ Function in daily living. ^4^ Function in sports and recreation. ^5^ Knee-related quality of life.

**Table 3 sensors-22-05289-t003:** ICC values and 95% confidence intervals for test–retest reliability of the measured accelerations obtained with SENS and Criterion.

Variable	SENS	Criterion
acc_x_	0.94 [0.93–0.94]	0.97 [096–0.97]
acc_y_	0.93 [0.93–0.94]	0.96 [0.95–0.96]
acc_z_	0.96 [0.95–0.96]	0.98 [0.98–0.98]
n	0.94 [0.93–0.94]	0.97 [0.97–0.97]

**Table 4 sensors-22-05289-t004:** Correlation and similarity-analysis results between SENS and Criterion signals in the time domain.

Variable	Side	CCC ^1^	NSI ^2^
acc_x_	Right	0.90 [0.87–0.92]	0.94 [0.92–0.96]
	Left	0.91 [0.88–0.93]	0.94 [0.93–0.95]
acc_y_	Right	0.89 [0.88–0.90]	0.96 [0.94–0.97]
	Left	0.90 [0.89–0.92]	0.96 [0.95–0.97]
acc_z_	Right	0.81 [0.80–0.83]	0.84 [0.73–0.89]
	Left	0.83 [0.82–0.84]	0.80 [0.72–0.86]
n	Right	0.90 [0.89–0.91]	0.95 [0.93–0.96]
	Left	0.90 [0.90–0.91]	0.96 [0.94–0.97]

^1^ Concordance correlation coefficient. ^2^ Normalized similarity index.

**Table 5 sensors-22-05289-t005:** Bland–Altman LoAs and biases for SENS measurements compared with Criterion (95% confidence intervals are provided in square brackets).

Variable	Side	Lower LoA ^1^	Upper LoA ^2^	Bias
**acc_x_**	Right	−0.18 [−0.20–−0.17]	0.22 [0.21–0.23]	−0.007 [−0.01–−0.003]
	Left	−0.17 [−0.18–−0.16]	0.21 [0.19–0.23]	−0.004 [−0.007–−0.0006]
**acc_y_**	Right	−0.34 [−0.37–−0.32]	0.34 [0.31–0.37]	−0.008 [−0.01–−0.003]
	Left	−0.32 [−0.35–−0.30]	0.32 [0.31–0.34]	0.001 [−0.003–0.007]
**acc_z_**	Right	−0.21 [−0.22–−0.19]	0.24 [0.22–0.28]	−0.005 [−0.009–−0.002]
	Left	−0.21 [−0.22–−0.19]	0.24 [0.23–0.27]	−wew0.005 [−0.006–0.004]
**n**	Right	−0.23 [−0.25–−0.22]	0.21 [0.20–0.22]	0.008 [0.004–0.012]
	Left	−0.23 [−0.24–−0.21]	0.22 [0.20–0.23]	0.007 [0.004–0.009]

^1^ Lower limit of agreement. ^2^ Upper limit of agreement.

**Table 6 sensors-22-05289-t006:** Correlations of the frequencies and powers of the peaks of the PSDs and the Fourier coefficients between SENS and Criterion measured by concordance correlation coefficients (CCCs).

Variable	Axis	CCC ^1^
Frequency of PSD peaks	*x*	0.99 [0.99–0.99]
	*y*	0.98 [0.97–0.98]
	*z*	0.99 [0.98–0.99]
	*n*	0.99 [0.99–0.99]
Power of PSD peaks	*x*	0.91 [0.90–0.93]
	*y*	0.87 [0.84–0.89]
	*z*	0.43 [0.34–0.52]
	*n*	0.87 [0.85–0.89]
Fourier coefficient	*x*	0.98 [0.98–0.99]
	*y*	0.92 [0.91–0.93]
	*z*	0.86 [0.85–0.87]
	*n*	0.99 [0.99–0.99]

^1^ Concordance correlation coefficient.

**Table 7 sensors-22-05289-t007:** Correlation and agreement of the frequencies and powers of the PSD peaks and the Fourier coefficients between SENS and Criterion.

	Axis	Lower LoA ^1^	Upper LoA ^2^	Bias
Frequency of PSD peaks	*x*	0.00 [−0.06–0.00]	0.17 [0.00–0.35]	0.00 [0.00–0.00]
	*y*	−0.13 [−0.59–0.00]	0.61 [0.27–0.84]	0.00 [0.00–0.00]
	*z*	−0.23 [−0.46–−0.08]	0.36 [0.00–0.70]	0.00 [0.00–0.00]
	*n*	0.00 [−0.29–0.00]	0.28 [0.17–0.35]	0.00 [0.00–0.00]
Power of PSD peaks	*x*	−0.20 [−0.22–−0.17]	0.25 [0.19–0.29]	0.005 [−0.004–0.02]
	*y*	−0.19 [−0.23–−0.15]	0.33 [0.22–0.38]	0.008 [0.001–0.02]
	*z*	−0.41 [−0.48–−0.31]	0.45 [0.38–0.53]	0.002 [−0.02–0.02]
	*n*	−0.21 [−0.22–−0.20]	0.34 [0.28–0.40]	−0.004 [−0.02–0.005]
Fourier coefficient	*x*	−0.05 [−0.06–−0.05]	0.05 [0.05–0.06]	−0.002 [−0.003–−0.001]
	*y*	−0.08 [−0.09–−0.07]	0.07 [0.06–0.08]	0.0002 [−0.0007–0.001]
	*z*	−0.06 [−0.06–−0.05]	0.05 [0.04–0.06]	−0.0005 [−0.001–0.0005]
	*n*	−0.06 [−0.06–−0.05]	0.05 [0.04–0.05]	−0.001 [−0.003–−0.0003]

^1^ Lower limit of agreement. ^2^ Upper limit of agreement.

## Data Availability

The data presented in this study are available upon request from the corresponding author.

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
