# Peer review of "Criterion Validity of Linear Accelerations Measured with Low-Sampling-Frequency Accelerometers during Overground Walking in Elderly Patients with Knee Osteoarthritis"

_sensors, 2022, doi:10.3390/s22145289_

Round 1

Reviewer 1 Report

This study aimed to determine the validity and reliability of low sampling frequency accelerometer measurements in KOA patients.  I have the following major suggestions.

1.    What is the novelty of this study although several low sampling studies have been reported earlier for KOA Patients?

2.    Please write down the contribution of the study at the end part of the Introduction section in bulleted form.

3.    Authors should include figures of experimental scenarios during the simultaneous recording of two sensor systems. Sensor positions of XSens systems should be described. The authors should add a figure of the experimental protocol used in this study.

4.    Authors may extend the background studies. Authors should review gait changes due to diseases and improve references by mentioning studies of various neuromuscular changes in the article, and prediction of myoelectric biomarkers in post-stroke gait.

5.    KOA patient characteristics should be reported in detail, such as KL grade, WOMAC grade or others.

6.    Authors should report simultaneous waveforms of two sensor systems.

7.     Abnormal gait is the consequence of diseases, such as stroke. Authors should review gait changes due to stroke in the article, real-time gait monitoring system for consumer stroke prediction service.

8.    How did the authors normalize the waveforms? Authors should report the gait cycle (0~100) waveforms of two sensor systems.

9.    Authors should include conceptual figures of their proposed method with more details and parametrization.

10.  The results and discussion section need to be extended and improved. Authors should discuss the strength and weaknesses of reported findings with other previous findings and improve the manuscript by adding a table in the discussion section.

Reviewer 2 Report

This study validated the accuracy of commercial IMU, low sampling frequency, for overground walking in elderly patients with knee osteoarthritis. As a result of the 44 patient’s experiments, it was confirmed that the low sampling IMU can be used for gait analysis. Unfortunately, it seems that the manuscript will not be enough for publication.

1)     The authors compared the measured acceleration values ​​of two different commercial products (SENs vs Xsens). Why compare the acceleration measurements between the two products? Does it make sense to simply compare the accelerometer accuracy of a device?? 

2)     Many studies on gait and activity analysis using Sens Motions product have already been published. What is the novelty of this study? 

-  Pedersen BS et al., “Validation of Two Activity Monitors in Slow and Fast Walking Hospitalized Patients,” Rehabil Res Pract. 2022.

- Bartholdy C, et al., “Reliability and Construct Validity of the SENS Motion® Activity Measurement System as a Tool to Detect Sedentary Behaviour in Patients with Knee Osteoarthritis,” Arthritis. 2018.

3)     As the author said, IMU sensor generally has the advantage that the battery can be used for a long time as the sampling rate is lower, but has the disadvantage of lower accuracy. In particular, in the case of using a low sampling rate, the accuracy is greatly affected by the speed of the activity to be measured. Therefore, many studies have been conducted to increase the accuracy at a low sampling rate of IMU.

- Sung et al., “Prediction of Lower Extremity Multi-Joint Angles during Overground Walking by Using a Single IMU with a Low Frequency Based on an LSTM Recurrent Neural Network,” Sensors, 2022.

- J. Liang et al., "Accurate Estimation of Gait Altitude Using One Wearable IMU Sensor," 2018 IEEE 1st International Conference on Micro/Nano Sensors for AI, Healthcare, and Robotics (NSENS), 2018, pp. 64-67

4)     As mentioned above, an IMU with a low sampling rate can be used only in very slow activity. What was the patient's average walking speed? How many total gait cycles did you compare?

5)     In Figure 2, the gait appears to occur once per second. Is that right?

6)     Unify the use of abbreviations.

Round 2

Reviewer 1 Report

Figure quality needs to be improved. 

Reviewer 2 Report

The author has put a lot of effort into improving the quality of the manuscript. However, important issues have not yet been resolved.

1) The purpose of motion analysis studies using IMU is to confirm the usefulness of the clinical application. Therefore, many studies have been conducted to classify physical activities or analyze gait using IMU sensors. As the author also said, comparing the acceleration values ​​of two devices is a very fundamental study. The procedures performed (low-pass filtering, interpolation, resampling…) to compare the accelerations of the two devices were included in the preprocessing process in most studies. Maybe, the reason the author considered it as the first study is that it may be a comparison of the simple preprocessing results. 

2) As mentioned above, it is necessary to derive a result suitable for the purpose of clinical use based on the advantage of using a low-sampling IMU. If the sensor device is normal, the acceleration value should be reliable for the same motion. 

3) Despite being an OA patient, the cadence is fast. If the average is 110 cadences, there are only 11.45 samples per step. In general, the number of acceleration samples of 11.45 per step is very insufficient for gait analysis.

4) Was there any difference in the measurements on the left and right legs?  

5) Did the author do anything to compensate for the disadvantages of low sampling IMU? If so, that explanation and emphasis are needed.

6) Since the study was conducted on OA patients, it is highly recommended to derive clinical significance from the IMU data. Or, it would be recommend to present a methodology to overcome the limitations of low sample numbers.

Round 3

Reviewer 2 Report

1)     Accuracy, precision, reliability, etc. have often been used to describe the results of the analysis. It is necessary to check whether these words are correctly used as an analytical index in the results.

2)     Modify Figure 1 to show all 17 Xsens sensor locations.

3)     Some English expressions need to be corrected, and there are unnecessary italics in the text.
